# Recent Advances in Adventitious Root Formation in Chestnut

**DOI:** 10.3390/plants9111543

**Published:** 2020-11-11

**Authors:** Jesús M. Vielba, Nieves Vidal, M. Carmen San José, Saleta Rico, Conchi Sánchez

**Affiliations:** Department of Plant Physiology, Instituto de Investigaciones Agrobiológicas de Galicia, Consejo Superior de Investigaciones Científicas, 15705 Santiago de Compostela, Spain; jmvielba@iiag.csic.es (J.M.V.); nieves@iiag.csic.es (N.V.); sanjose@iiag.csic.es (M.C.S.J.); saleta@iiag.csic.es (S.R.)

**Keywords:** adventitious rooting, auxin, *Castanea*, maturation, microshoots, trees

## Abstract

The genus *Castanea* includes several tree species that are relevant because of their geographical extension and their multipurpose character, that includes nut and timber production. However, commercial exploitation of the trees is hindered by several factors, particularly by their limited regeneration ability. Regardless of recent advances, there exists a serious limitation for the propagation of elite genotypes of chestnut due to decline of rooting ability as the tree ages. In the present review, we summarize the research developed in this genus during the last three decades concerning the formation of adventitious roots (ARs). Focusing on cuttings and in vitro microshoots, we gather the information available on several species, particularly *C. sativa*, *C. dentata* and the hybrid *C.*
*sativa* × *C. crenata*, and analyze the influence of several factors on the achievements of the applied protocols, including genotype, auxin treatment, light regime and rooting media. We also pay attention to the acclimation phase, as well as compile the information available about biochemical and molecular related aspects. Furthermore, we considerate promising biotechnological approaches that might enable the improvement of the current protocols.

## 1. Introduction

The genus *Castanea* belongs to the *Fagaceae* family and includes seven species of deciduous trees and shrubs [1]. *Castanea* spp. trees are members of a genus with great interest, due to their geographical extension, ecological relevance and different human services, that include timber, nut production and other uses [2], highlighting their multipurpose character. European chestnut (*Castanea sativa* Mill.) is a major tree species covering more than 2.5 million hectares, mainly in countries around the Mediterranean basin [3]. The American chestnut [(*Castanea dentata* Marsh) Borkh.] suffered an enormous decline in the 20th century due to the chestnut blight canker disease caused by the fungus *Cryphonectria parasitica* [(Murril)Bar], as well as by the ink disease produced by the oomycete *Phytophthora cinnamomi* (Rands). European chestnut has also been extensively affected by *Phytophthora* spp. causing high mortality of trees in natural stands. Chinese chestnut (*C. mollissima* Blume) is responsible for more than 80% of the world chestnuts production, and its cultivation is extensive in China and other Asian countries. Nonetheless, they produce nuts of poorer quality than European chestnut. The Japanese chestnut (*C. crenata* Sieb and Zucc.) is a small tree native of Japan, where it is widely spread. Both Asian species are ink- and blight canker-resistant, but at different levels. This natural resistance has been used for more than a century for the generation of hybrid trees that could carry the useful traits of the menaced species, but at the same time be able to resist the presence of those pathogens [1]. Moreover, plant improvement, including biotechnological approaches in this genus, have been focused on the generation of tolerant/resistant trees [4,5]. On the other hand, the Asian species have long suffered the effects of the chestnut gall wasp (*Dryocosmus kuriphilus*), an insect that can cause severe damage and death of the infested trees. This insect has also become a menace for the European and American chestnuts, as it has been introduced in their planting areas in the last decades, probably through the introduction of Asian-origin material for the generation of hybrids. Resistance to chestnut gall wasp has not been identified so far in *C. sativa*, but it has in other *Castanea* species [6].

Vegetative propagation of chestnut is still difficult to achieve because conventional methods like grafting are time-consuming or unsuccessful due to incompatibility between the scions and the rootstocks [7]. Although rooting of cuttings is an essential methodology in the clonal propagation of selected genotypes, practical approaches based on de novo root formation are limited by the poor rooting ability of these species, since chestnuts are considered difficult-to-root [8,9]. Moreover, there is an early decline in the rooting potential of cuttings with increasing age of the mother plant. The maturation-related loss of rooting competence seems to occur rapidly during plant development [10], but the molecular mechanisms behind this process are not defined. Micropropagation of material that retains juvenile characteristics (basal shoots, stump sprouts) appears to be the most effective approach for clonal propagation of elite genotypes that can only be selected during their adult phase [11,12]. However, due to its long history of seedling cultivation and hybridization, the species of *Castanea* have produced many local genotypes and great heterozygosity [1], including a genotype-dependent rooting response. This behavior hinders the establishment of general protocols for the genus, and specific research is required for every species, and generally for every genotype. With this troublesome background, efforts have moved towards the optimization of the existing protocols for both juvenile and mature cuttings. Besides recent progress, commercial vegetative propagation of chestnut species is still a bottleneck for the forest industry.

In the present review, we analyze the achievements in the formation of adventitious roots (ARs) in chestnut species and hybrids, with greater emphasis in the European and American species because most of research related to adventitious rooting (AR) has been done in these species. The production of ARs in chestnut is an essential step in the propagation of selected genotypes which show interesting market-related traits, like high-quality nuts or biomass production, or for the multiplication of trees that show resistance to the specific diseases mentioned. We mainly focus on ARs developed from stem cuttings after auxin treatment, which generally initiate from cells neighboring vascular tissues, mostly cambium and parenchyma cells [13]. The origin of spontaneous roots developed in auxin untreated shoots that were maintained in proliferation medium was also investigated. We also take into account the biochemical and molecular findings in these species that might pave the way for biotechnological progresses.

## 2. Current Knowledge on Adventitious Root Formation

The ability of plants to form ARs is an essential developmental trait in plant biology. It is a complex physiological and molecular process that allows the generation of roots from tissues other than the primary root (mainly stems and leaves), whether constitutively or in response to different types of stress such as wounding and flooding [14]. This somatic developmental plasticity of plants enables them to adapt to changes in their environment, respond to stressful situations and recover from traumatic damages. Due to its relevance for vegetative propagation, AR has been the focus of extensive research in monocot and dicot model species, with recent reviews providing current state-of-the-art information [14,15,16]. A significant knowledge of the physiological and molecular basis of the process has been collected, however several questions still need to be addressed. For instance, its similarity with primary and lateral root formation, the influence of genetic variability or the extent to which knowledge in some species can be transferred to others [15,16,17]. Furthermore, analysis of the AR formation in woody cuttings lags behind that of model species, as other factors such as maturation and recalcitrance to regeneration, or functional limitations like the lack of mutant phenotypes and proper transformation and regeneration protocols hamper the analysis of AR formation in these species [18]. Nonetheless, recent reviews have gathered the available information for cuttings [17,18,19], pinpointing the complex interaction between hormones and their cross-talk (mainly auxins, jasmonic acid and ethylene), the nutritional/metabolic state and the epigenetics profile. Recalcitrance of tissues in terms of rooting competence is a major limitation in the clonal propagation of many species, and it is deeply related to their maturation state [20].

ARs initiate from a variety of somatic cell types exhibiting distinct degrees of differentiation, plasticity, or competence for root regeneration, depending on the species, the explant type, the maturation stage or the inductive stimuli, among other factors. The process of AR is divided into three main phases that seem to be conserved in every species analyzed [15,21,22]. In general terms, tissues forming ARs go sequentially through three developmental stages: induction, initiation and expression. Hormonal requirements and morphological changes in each phase are similar in most species studied so far. During the induction phase specific target cells undergo cellular reprogramming, acquire rooting competence, and initiate a root developmental program in response to the root induction stimuli. If the target cells are root-competent (i.e., cambium cells) they can lead directly to the formation of root initials in the presence of an inductive factor [23]. However, in many woody species, ARs initiate from cells that show a certain degree of differentiation and from cells that are not pre-specified for this root developmental pathway. In that case, an early dedifferentiation phase taking place before [20,21,22] or at the early stages of the induction phase [23] has been proposed. Dedifferentiation is the process whereby the responsive cells revert to their previous status [24] or to a stem cell-like state [25] and, eventually, increase their developmental potential [26]. In this sense, somatic embryogenesis could be considered a genuine dedifferentiation process [24]. However, whether the progression of differentiated somatic cells towards a less differentiated state is a real dedifferentiation or just a transdifferentiation is still a matter of debate [24]. Indeed, some authors suggest that both dedifferentiation and transdifferentiation are synonymous terms referring to the enhanced developmental potency of the cells, and that cellular reprogramming might be a more accurate description [26]. This cell reprogramming might be activated by stress (i.e., wounding), leading cells to free from their fate restriction [27]. According to the model proposed by Da Costa et al. [23], wounding is the initial event of the AR process in cuttings, triggering hormonal changes, and the subsequent activity of the auxin transport machinery would lead to the asymmetric accumulation of this hormone between neighboring tissues. By these means, the proper signaling cascades area activated by auxin in the cells where the hormone concentration reach a peak, and these cells switch their genetic program to the formation of ARs. It is generally accepted that both wounding and exogenous auxin are required for AR in cuttings [23], and results in chestnut support this idea (see below). The establishment of a localized auxin maxima suggests a significant role for the auxin transport proteins, that might be the prevailing constituent governing the cellular origin of the new root. Asymmetric accumulation of auxin between neighboring tissues would be the required signal to activate cell proliferation [28]. At the initiation phase, root initial cells begin to divide and organize into a dome-shaped structure that will become the meristemoid. For the correct founding of the polarity of the new organ, a reorientation of the cell division plane is necessary [18], emphasizing the potential role of microtubules and cell-wall remodeling enzymes in the development of ARs [19]. During the expression stage, the autonomous new organ grows across the different tissues of the plant towards the epidermis, and finally emerges from the stem [22]. It is generally accepted that a transient peak of endogenous auxin levels determines the end of the induction phase and the beginning of the initiation phase [29]. Whereas low auxin levels are required during the initiation phase, those levels increase again during the expression phase [29]. Recent works are focusing their attention on the roles played by other hormones, as well as their intricate interactions with auxin in every phase of the process [13,19].

## 3. Developmental Phases of Adventitious Root Organogenesis in Chestnut: Histological Analysis

As previously mentioned, the developmental phases of AR have been described in apple cuttings [22] and similar sequential phases were described in other species [30,31]. In this section we will mainly focus on the anatomical studies performed during the formation of auxin induced ARs in chestnut microshoots. Besides, some information is also provided on the origin of spontaneous roots developed in auxin untreated shoots.

In *C. sativa*, juvenile-like microshoots (BS; derived from basal shoots) and adult microshoots (C, derived from crown branches) were used in the analysis performed by Ballester et al. [32]. As shown in Figure 1, in auxin treated BS microshoots ARs initiated from cells belonging to the cambium or neighboring zones, including xylem or phloem parenchyma cells [32,33], as reported in other species.

During the induction period, certain cells in the cambium and neighboring areas are more densely stained and their nuclei and nucleoli become more pronounced (Figure 1a–c) [32]. Similar characteristic features are also found in other related species, like *Quercus robur* [34]. These histological traits might be related to a more intense transcriptomic activity connected to a cellular reprogramming. The specific cellular origin of the new ARs could be a consequence of the integration of the molecular signaling cascades activated mainly by wounding and auxin, but also precise positional cues are essential for the specification of root initials. The differential auxin distribution between the cells of tissues is essential to modify the developmental program of the target cells that will become root initials and drive the generation of the new root [17]. The generation of this auxin gradients suggests a significant role for auxin transport proteins (PIN, AUX) in the process. Periclinal and anticlinal mitotic divisions are already detected after 24 h of indole-3-butyric acid (IBA) treatment in the cambial zone (Figure 1a). The number of cell divisions increased from 24 h onwards, not only in the cambium, but also in other tissues, particularly in the ray parenchyma and phloem region (Figure 1b,c). Meristemoids formed by small clumps of densely stained isodiametric cells with meristematic characteristics are observed after 3–4 days of auxin treatment (Figure 1d) as previously reported [32,33]. The patterning of the new organ is acquired by an asymmetric division that involves a change in the cell division plane [19,32]. After 6 days, some meristemoids become true root primordia (Figure 1e), that grow through the surrounding layers of tissue and develop a vascular system connected to that of the stem during the expression phase (Figure 1f) [32,33,35]. Different stages of root development are observed in the same section as the ARs formation is an asynchronous process. It has been proposed that asynchrony might be related to the extent of specific phases, particularly induction [36]. Molecular mechanisms involved in the emergence of the new ARs have not been explored in chestnut, but it is believed that they mainly resemble those of the lateral roots in other species. Auxin activity modulates the hydraulic properties and the walls of the cells surrounding the root primordia by controlling the expression of its own transporters [37]. Nonetheless, the activity of auxin in this stage must be strictly controlled in order to preserve the integrity of those surrounding tissues, as seen for lateral roots [38]. Finally, primordia start their outgrowth from the stems around day 10 after the beginning of the process. Nonetheless, this outgrowth can be detected sooner in leaves and petioles, probably because the root primordia need to grow across a lower number of tissue layers [39,40]. Indeed, root induction in leaves also seems to present a greater level of synchrony in root emergence [40]. Synchronous development of roots is also of commercial interest, as it helps in the management of plants in breeding programs.

On the other hand, in C non-rooting competent shoots, the progression of cell divisions initiated in the cambial derivatives, especially on the phloem side, failed to organize properly into a meristemoid but rather gave rise to a callus structure [32]. Different auxin-derived calli have been analyzed at the transcriptomic level in Arabidopsis, showing limited overlap in gene expression between the different types of calli obtained from different tissues and under distinct conditions [26], with some of them sharing components of root developmental pathways [41]. Thus, the callus generated in mature chestnut shoots, with the same cellular origin and under identical treatment as root-forming juvenile shoots, may be expressing similar genetic routes as root-forming shoots. However, other physiological or epigenetic factors impede the establishment of the root developmental program as those cells do not acquire the rooting competence. The inability of the mature tissues in chestnut to generate root initials might be related to their failure in establishing a proper auxin gradient, rather than in the inability to respond to the hormone. A major shift in the homeostasis of auxin as plants mature has been proposed previously [42], and the results in chestnut sustain this idea [43].

Although with low frequency, ARs can be developed in BS chestnut shoots maintained for more than two months in the proliferation medium without the application of exogenous auxin treatment. The formation of spontaneous roots has been described in cuttings from some woody species as *Populus* [23,44]. As shown in Figure 2a,d, shoots have an aged appearance, with thick stems, apical necrosis symptoms, and senescent leaves. At first glance, these roots seem to originate from the callus developed at the base of the shoot. Nonetheless, after removing the surrounding callus, it has been observed that they arise from the stem (Figure 2b,c). The histological analysis was not sequentially performed, but only when roots emerged from the callus, showing the callus tissue surrounding the stem base (Figure 2e), as well as the emergent root (Figure 2h).

The analysis revealed that new roots arise from the stem tissue (Figure 2f,g) and they probably have been initiated from the cambial zone or parenchyma tissue. The vascular tissue of the new root is connected with the stem vascular tissue (Figure 2g). Therefore, it appears that the AR initiation and callus formation could share initial genetic pathways as reported by Liu et al. [45] during de novo root regeneration in leaf explants, but then followed a different development pattern. Indeed, it has been shown in Eucalyptus that roots can be formed directly from callus [46] suggesting a complex relation between both types of organs. Our hypothesis is that auxin accumulation at the wounding zone induces the genetic reprogramming of certain cells in the stem base that somehow become competent cells. Later, a new auxin gradient is created by other stimuli, like the stress generated by the deficit of the mineral nutrients in the media, as well as changes in the hormone balance at the base of the shoots, as those cells acquire the root initials fate and establish the root signaling pathway. Similarly, the generation of auxin gradients seems to be involved in the development of actinorhizal nodules in *Casuarina glauca* and *Discaria trinervis* [47,48]. Therefore, an in-depth analysis should be performed to define this process. In some woody species, ARs are formed spontaneously without auxin treatment, however this process is not common in chestnut.

## 4. Biochemical Profiles during AR

### 4.1. Auxin Content

Auxin is known to be the master regulator of the process of AR [17]. Auxin homeostasis within the plant is affected by several factors, including synthesis, transport, conjugation and degradation. Since IBA is the most common compound used for the induction of ARs, it has also been analyzed to understand its role within the process. It is important to recall that, so far, IBA is believed to act through its conversion to IAA (indole-3-acetic acid) [49], though some results suggest that in specific responses, IBA itself displays auxin activity [50].

In *C. sativa*, the endogenous auxin dynamics in the juvenile-like and adult microshoots treated with a pulse of 4.9 mM IBA was analyzed previously [32]. The results showed that the IAA content was higher in non-rooting mature tissues (about 5 times) than in juvenile-like shoots, while the IBA content was quite similar in both types of shoots. These results contrast with those reported in *Eucalyptus* spp. [51,52,53] and Arabidopsis (reviewed in [16]), in which a poor rooting ability was directly correlated to a low auxin content. As shown in Figure 3, similar results to those previously reported [32] were found when shoots of juvenile-like and mature characteristics were treated with 125 µM IBA for 24 h. The analysis was performed separately in the basal (Figure 3a) and apical sections of the shoots (Figure 3b), treated or non-treated with auxin.

The IAA content was very low in auxin untreated materials (which do not root) regardless of the analyzed section, and between 5–10 times lower in apical sections than in basal sections in treated shoots. After auxin treatment, IAA levels were higher in rooting incompetent mature tissues than in the competent juvenile-like tissues, particularly in the basal sections (Figure 3a). The peak of IAA detected after 24 h of auxin treatment could be associated with the activation of cell divisions, but the lack of rooting ability of mature shoots cannot be explained by their total endogenous content. Gonçalves et al. [54] studied the auxin content in micropropagated shoots derived from an easy-to-root rootstock of hybrid chestnut (*C. sativa* × *C. crenata*). These shoots were subjected to two IBA treatments, “dipping” and 24 h, with more than a 90% rooting in both cases. As reported in European chestnut, an increase of IAA levels was detected in response to IBA, with a peak of IAA after 24–48 h of root induction depending on the treatment. Similarly, a positive relationship between auxin content and rooting was reported with *C. mollissima* material [55]. In this study, both the rooting ability of microshoots and their IAA levels after IBA treatment increased with the number of subcultures.

Even if it seems that an IAA increase during the first hours of rooting induction is a requisite for obtaining a successful rooting response, the current data cannot explain the difficulties to root exhibited by other materials in which IAA levels also increased with IBA treatment, as happened with the shoots derived from crown branches of many adult trees. It must be kept in mind that the auxin profiles shown previously were obtained from a pool of many tissues at whole-stem level, and may not represent the IAA content of the cell type or tissues closely related to the rooting process. Techniques that allow for single-cell resolution, or the use of laser micro-dissection of the tissues involved could help achieve more accurate results in the quantification of auxins.

### 4.2. Polyphenols and Polyamines

Polyamines are low molecular weight cations that have long been studied in order to clarify their role in the rooting process. Their mode of action is not clear, but it is suggested that, among other processes, they are involved in cellular division and differentiation through diverse mechanisms, including interaction with kinases and transcription factors [56]. The major polyamines in plants, putrescine, spermidine and spermine, were analyzed in juvenile-like and adult chestnut microshoots after IBA treatment [32]. Mature tissues showed a greater content in polyamines, suggesting a negative relationship with the induction of ARs [32]. Nonetheless, recent research has shown that exogenous application of spermidine promoted ARs formation in *Malus prunifolia* stem cuttings, affecting auxin and gibberellin homeostasis [57]. Due to the high number of processes in which polyamines seem to take part, it is difficult to clearly establish their role in AR. Besides, the activity of their catabolic enzymes and the concomitant generation of reactive oxygen species add another layer of complexity to their putative role in this developmental process [58].

Polyphenols are believed to enhance rooting by protecting auxin against oxidation or by modulating its transport, in the case of flavonoids ([59], and references therein). Methyl gallate, more abundant in juvenile than in adult cuttings of hybrid chestnut, was found to exert a significant protective role on IAA against oxidation [60]. Phenolic content in juvenile-like and mature shoots of the same genotype was analyzed in order to identify positive or negative relationships of specific compounds with the AR process [61]. Differences in the content of certain compounds (ellagic tannins, flavonols) between juvenile and mature tissues were detected from the beginning of the experiment, but no unambiguous relationships with the ability to root could be established [61] Furthermore, previous results had suggested that genotype variability might prevent extrapolation of the results obtained for any of those compounds [62,63].

Long cuttings of hybrid chestnut were shown to root better than the short ones, and the content of flavonols and ellagic acid in the leaves was found to be directly related with this trait [64]. With this same working system, Osterc et al. [65] also found a positive relationship between the content of rutin in the leaves of the cuttings and their ability to root. Despite these interesting results, the diversity of polyphenolic compounds makes it difficult to establish accurate relationships with AR and other processes.

### 4.3. Auxin-Related Enzymatic Activity

We were only able to find two reports dealing with the analysis of auxin-related enzymatic activity during AR in chestnut, once again highlighting the lack of reliable knowledge in specific areas of this process. Nonetheless, these studies might not be conclusive, as the putative existence of isoenzymatic forms or several closely related genes can render the analysis incomplete.

Peroxidase activity is believed to be related to IAA-oxidase catabolism, and therefore might be a good marker of auxin metabolism during AR. This activity was measured in hybrid chestnut under two auxin treatments and in control shoots, but no significant differences were found within the first 48 h after the beginning of the experiment, suggesting the lack of a relationship of this enzyme with the early events of AR [33]. In *C. mollissima*, auxin content in shoots was shown to increase with the number of subcultures from the beginning of the establishment of the plants in vitro, as previously mentioned [55]. Authors measured different enzymatic activities related to IAA metabolism in the same material, showing an increase of polyphenol oxidase activity concomitant with the increase of auxin, but at the same time they found a decrease in the activities of IAA-oxidase and peroxidase-oxidase [55], negatively linking the activity of the two latter enzymes with formation of ARs. In Arabidopsis, it is believed that the major auxin oxidation pathways include the activity of the enzyme dioxygenase for auxin oxidation 1/2 (AtDAO1/2), with the corresponding mutant showing a decrease in the presence of auxin oxidized metabolites [66]. These oxidative pathways act in a coordinated fashion with the auxin conjugation routes, regulating active auxin content in the tissues [67]. No DAO activity has been measured in chestnut, but some auxin-conjugating genes have been analyzed (see below). More work is needed in this field to clearly link the action of these enzymes with specific processes in chestnut rooting.

## 5. Molecular Aspects of AR in Chestnut

### 5.1. Genetics

At the genetics level, AR formation in chestnut cuttings is believed to be under moderate additive genetic control [10], however other factors like maturation or auxin homeostasis are relevant. Moderate inheritability of AR traits has also been suggested for other woody species, like *Populus* [68].

Most of the studies have been carried out at the physiological and biochemical levels, and no complementary molecular and genetic studies have been developed. Most of the efforts in this sense come from the work developed in our laboratory. Due to the nature of our working system, with juvenile-like and mature lines of the same genotype, some of the results might also be related to other developmental processes. The expression of the *CsCPE* gene, that codes for a glycine-rich small polypeptide, was analyzed in both types of lines, showing greater expression in mature tissues [69]. Further analysis of the gene expression during the induction and development of somatic embryos in chestnut and oak [70,71] suggest a role of the gene in the regeneration ability of these recalcitrant species.

Five partial sequences showing differential expression patterns in juvenile-like and mature tissues in response to auxin were isolated by differential display and designated *Cs114F2A*, *Cs312A0C*, *Cs914F2G*, *Cs410A0C* and *Cs714F2G* [72]. Recent homology searches have shown that three of them belong to defense-responsive genes (*Cs114F2A*: oxalate-CoA ligase; *Cs312A0C*: Metallothionein-like; *Cs914F2G*: Detoxification protein 49), while *Cs410A0C* shows homology with the α/β hydrolase superfamily, and *Cs714F2G* is a MYB61-like transcription factor. *Cs114F2A* showed a higher wound-responsive induction in rooting competent tissues than in mature tissues, while expression of *Cs312A0C* was greatly influenced by auxin in both types of tissue. The α/β hydrolase-like gene showed greater response to auxin in juvenile-like shoots, but lack of characterization of the activity of this putative enzyme prevents us from any conclusions. *Cs714F2G* (MYB-like 61) was induced by wounding and auxin in both tissues as early as 1 h after treatment. This result is in agreement with findings in other species, where it has been shown to be expressed at sink tissues [73] such as the wound site in cuttings (reviewed in [17]). Significant expression of *Cs714F2G* was detected at 12 h only in juvenile shoots treated with auxin. Interestingly, MYB61 genes have also been shown to modulate root system architecture as they have effects on the activity of gibberellins [73,74]. Therefore, further exploration of the activity of this gene and its relationship with AR is necessary.

Expression analysis of other candidate genes isolated from chestnut was analyzed during the induction of ARs in juvenile-like and mature shoots. Generally, different expression patterns were detected in both types of shoots under the same conditions, suggesting that a major shift in gene expression takes place during maturation. *Castanea sativa Scarecrow-like 1(CsSCL-1)* gene codes a transcription factor from the GRAS family and is induced by auxin within the first 24 h after treatment [75]. Whereas localized expression in the cambium was detected in rooting-competent shoots, a spread signal through several tissues was observed in mature shoots [76]. Other related GRAS proteins have been shown to take part in the establishment of meristematic identity in root cells that will eventually form the ground tissues of the root, governing the asymmetric divisions of the cells from the quiescent center, giving rise to the new tissues [77]. Altogether, these data suggest a significant role for this gene in the formation of ARs. Furthermore, the expression in mature shoots confirms that both callus and AR formation share elements in their genetic pathways [41]. The expression analysis of a gene from the GH3 family, in which members modulate the availability of free auxin by conjugating amino acids to IAA [78] was also characterized during the AR in both types of chestnut shoots [43]. As expected, *CsGH3-1* was highly induced by auxin, but with different patterns in juvenile-like and mature shoots. Induction occurred earlier in juvenile shoots, suggesting a more readily gene responsive machinery for auxin in these shoots, while in mature shoots the induction lasted longer. On the other hand, the higher expression levels detected in mature than in juvenile-like shoots, suggests a greater activity of the protein that lowers the available free auxin in mature tissues [43]. In contrast, levels of the *GH3.3* transcripts in cuttings of *Arabis alpina* were associated with the rooting ability of cuttings from different genotypes, but not with the endogenous levels of IAA [79]. Therefore, as there are more active GH3 genes and their activity must be taken into account, it can be concluded from the *CsGH3-1* expression that, apparently, homeostasis of auxin changes with the maturation process, affecting AR, among other processes. Indeed, we are currently analyzing the expression of some proteins related to auxin transport, like PIN proteins. Preliminary results show differential expression of a PIN-like gene in auxin-treated shoots of the same genotype, but different ontogenetic state, with greater levels of PIN transcript in juvenile-like tissues (unpublished results). Though these data must be taken cautiously, they suggest greater activity of polar auxin transport proteins in rooting competent tissues, which might be involved in the establishment of the proper auxin gradients within shoot tissues, putatively concentrating auxin maxima in the cambium or neighboring responsive cells. In *Eucalyptus grandis*, which is an easy-to-root species, the influx and efflux auxin carrier proteins where shown to be induced by auxin [52]. The activity of these proteins might be directly related to the formation of auxin gradients between neighboring tissues, which is necessary for tissues patterning and morphogenesis [23].

Nonetheless, expression of *CsSCL-1* and *CsGH3-1* does not seem to relate to wounding signals, as control shoots did not show significant expression [43,75]. Therefore, other signaling pathways must integrate the information derived from this stress, which involves activity of reactive oxygen species and jasmonic acid [14]. In a recent work, we analyzed the expression of an ERF transcription factor (*CsRap2.12 like-1*) in the same working system as mentioned above. Results showed a slight induction of the gene at 6 h after wounding in both control and IBA treated cuttings, while IBA plus wounding induction was also clearly detected at 12 h by in situ analysis in the cambium zone of rooting competent tissues [80]. Indeed, at 12 h the expression pattern of this gene coincides with that of *CsSCL-1* [76]. It has been shown that genes from the GRAS and the ERF family can heterodimerize [81], and therefore it is tempting to speculate that both genes can work together in activating the specific gene expression cascades for the development of ARs, with *CsSCL-1* integrating the auxin signals, while *CsRap2.12 like-*1 would integrate the wounding signals. Nonetheless, more research is needed to support this hypothesis.

Overall, despite the significance of the findings about the activity of the genes mentioned, it is worth mentioning that some of them belong to multigene families, and the characterization of their related members could help clarify their specific role in AR.

### 5.2. Epigenetics

No analyses regarding epigenetic modifications have been carried out so far to analyze AR in chestnut. Due to the clear effect of maturation on the rooting ability, these analyses might provide relevant clues concerning the molecular basis of recalcitrance, as it is widely accepted that changes in gene expression in response to maturation are related to epigenetic changes (cytosine methylation, histone acetylation,) or the activity of microRNAs. It has been shown in Arabidopsis that the activity of miRNA156 and miRNA157 is directly linked with the vegetative phase change, through their effect on the expression of Squamosa Promoter Binding protein/SBP-like (SBP/SPL) transcription factors [82]. Indeed, overexpression of miRNA156 delayed the transition from the juvenile to the adult in a transgenic poplar tree [83]. Related work in Eucalyptus has shown the involvement of miRNA156 in the loss of rooting ability of the trees [84]. Moreover, different degrees of cytosine methylation of DNA were found between juvenile and mature tissues in chestnut [85]. It is believed that the methylation state is a stress-sensitive trait, which can be modulated by specific conditions [86]. Therefore, the development of protocols that can adjust the epigenetic state of mature tissues is an unexplored possibility. These protocols might provide the biotechnological tools that would help overcome the limitation in vegetative propagation imposed by maturation in these species.

## 6. Main Factors Influencing the Rooting Response

Although many facets described for model species can be validated in chestnut, its woody nature and established recalcitrant behavior impose some limitations to the applicability of these findings. Identification of the mechanisms underlying the elusive responses of these species will enable biotechnological tools to be applied, and improve their rooting performance, both in vitro and ex vitro. In this section we review most of the reports on rooting responses of chestnut materials published in the last three decades. Studies on AR on chestnut reported before 1990 are summarized in [1,4,87]. Major efforts have been focused in *C. sativa*, *C. dentata* and the hybrids *C. sativa* × *C. crenata*. The best results achieved in the occasional reports of AR performed on cuttings are summarized in Table 1. At optimal conditions established in each of these studies, longer cuttings performed better than the shorter ones [64,65]. Rooting frequencies affected by both auxin and genotype were reported in minicuttings of *Castanea* hybrids [88]. The genetic control on AR has been reported in juvenile chestnut cuttings of different full-sib families [10], in which both the presence and the number of roots traits were mainly controlled by non-additive variance.

The large amount of rooting research carried out with in vitro microshoots indicates the relevance of this methodology in the propagation of chestnut (Table 2).

In Vitro systems provide several advantages, including the availability of a greater number of homogenous shoots as well as the avoidance of environmental interferences. Nonetheless, the root systems generated might not always be optimal for acclimation and survival ex vitro (see below). We focus on the analysis of the following factors: genotype, auxin treatments, chronological age of the mother plant, maturation, ontogenetic stage, light, rooting media, among others.

### 6.1. Genotype

The genetic variability that exists within and among species of *Castanea* greatly influences the rooting performance of the trees. As shown in Table 2, large differences on rooting ability were observed among genotypes. A genotype-dependent rooting response was even noticed in juvenile microshoots established in vitro from embryonic axes that were isolated from nuts harvested from the same hybrid HV mother tree [90]. In BS microshoots of different clones subjected to the same treatment, there were large differences in their rooting percentages [90,93]. These reports clearly show the strong effect of genotype on the rooting performance of chestnut microcuttings.

Rooting frequencies seem to be more affected by genotype than by the method of auxin application since in three different clones (P1, P2 and 90025) these frequencies kept stable in microshoots subjected to different auxin inductive treatments [90,91,93]. Similarly, rooting percentages of leaves excised from microshoots exhibited similar rooting percentages as the donor microshoot [40]. However, within the same application method, it is important to define the appropriate IBA concentration for each genotype, particularly when roots are induced with high concentrations of IBA [92] as it can seriously affect shoot tip necrosis. Moreover, other parameters, like mean root number can be affected by the auxin treatment [33,91]. The genotype effect on rooting performance has been previously documented in other difficult-to-root species like walnut [103], oak [104,105], American beech [106] and holm oak [107]. In conclusion, besides other factors, genotype is critical in the formation of ARs in woody species.

### 6.2. Auxin Treatments

Information provided in Table 2 clearly shows that IBA is the preferred auxin for ARs induction in chestnut, as it has been reported in other woody species [34,108,109]. It has been suggested that IBA functions as an auxin precursor of IAA, the predominant form of active auxin in plant tissues, but it is more stable than IAA in culture media [110]. Although IBA is often detected at low levels in many plants and some evidences suggest that IBA itself lacks auxin activity [111] it has an important role in auxin homeostasis. It seems that the physiological effects of IBA are likely caused by IBA-to-IAA conversion. However, many questions regarding IBA metabolism, transport and signaling routes remain opened [49].

Two methods of auxin application are commonly used to induce ARs in chestnut: a short pulse of a high concentration of the hormone (sometimes referred as “dipping”), and the incubation of microshoots with a lower auxin concentration for a longer period (one to several days).

The application of exogenous auxin during the whole rooting period provided also high rooting percentages in seedlings derived microshoots [100,112]. These results are somehow surprising, as the continuous exposure to exogenous auxin might inhibit the initiation phase of AR [22]. Similarly, juvenile microshoots of *C. henryi* rooted well when they were subjected to the continuous exposure to 7.5 µM IBA, whereas no roots were formed when NAA (naphthalene-acetic acid) was used instead of IBA [100]. In cuttings of *Pinus caribea*, IBA treatment also proved to be more effective for rooting than NAA [113] whereas in Douglas fir stem cuttings specific concentrations of NAA (24.6 mM) or IBA (7.4 mM) provided similar percentages of rooting [114]. During the root regeneration from chestnut cotyledons, the presence of either NAA or IBA in the medium produced the highest number of roots [39].

The dipping method was used to induce roots in both cuttings and microshoots (Table 1 and Table 2), but at different range of IBA concentration and application period. Whereas cuttings were treated with concentrations of IBA between 12.5–49 mM for 2-10 s, the microshoots were generally dipped into the IBA solution (5–10 mM IBA) for a longer period (30–120 s). By these means, a significant amount of hormone enters the plant tissues through the cut surface of the shoots and will eventually enter the cells by pH trapping or the activity of influx carrier proteins (reviewed in [115]). In microshoots derived from stump sprouts/basal shoots, which retain juvenile characteristics, rooting frequencies were highly dependent on the genotype ranging from 19–97% (Table 2). Though this is a useful method, a stress signaling may be imposed by this auxin concentration, causing high levels of apical necrosis in microshoots. The presence of necrotic tissues and reduction on rooting has been reported in chestnut cuttings dipped in high concentration of IBA [89]. By using the incubation method, microshoots were induced to root with IBA concentrations several orders of magnitude lower (ranging from 2.5–250 µM) for an extended period of 24 h up to 28 days (Table 2). Hormone is included in the growing media, and it is absorbed in a slower and sustained fashion, allowing the plant to have a more controlled management of IBA, which can be directed towards the specific responsive cells when other factors concur. It has recently been shown that healthier plantlets were obtained with longer IBA treatments in comparison with the dipping method [91], which might be related to a less stressful situation for the shoots that prevents shoot-tip necrosis and helps in the formation of the roots at the wound site. In the clone P1, the rooting frequency as well as the number of roots was significantly improved by increasing the IBA concentration [91]. When both types of treatments have been applied on the same material at optimal auxin concentrations, results do not differ significantly with respect to rooting rates, though other traits like mean root number might be affected [33,91]. Therefore, though a genotype-dependent effect in the results for both types of treatments cannot be ruled out, it seems that optimization of the protocols might reduce such effect. Nonetheless, longer treatments for microshoots are labor-intensive, and it is a costly process that is not always suitable for commercial purposes.

### 6.3. Non-Auxin Compounds

Two urea derivatives, *N*,*N*′-bis-(2,3-methylenedioxyphenyl)urea (2,3-MDPU) and *N,N*′-bis-(3,4-methylenedioxyphenyl)urea (3,4-MDPU), have been shown to induce or facilitate the formation of ARs in several species [116]. The positive effect of these compounds on AR has been recently reviewed [117] and their effect on rooting of chestnut microshoots has also been tested [91]. Though their ability to induce AR was low and only visible when suboptimal auxin concentrations were used, they significantly improved other rooting-related parameters of the shoots, including the formation of the ARs at the wounding site, a greater number of lateral roots and reduced shoot tip necrosis of the shoots, among other factors analyzed [91]. Their mode of action is not clear, but the authors suggested a direct relation with an improved homeostasis of both auxin and cytokinin in the shoots [91].

Naphthyl-phthalamic acid (NPA) is an auxin transport inhibitor widely used for research purposes, whose specific mode of action is not yet defined unambiguously [118]. Its use can help unravel fundamental aspects of the rooting process by blocking the movement of IAA in the tissues. When NPA was applied alongside IBA to chestnut rooting-competent microshoots or leaf explants, there was a drastic decrease of rooting rates and a delay of 3–4 days on the emergence of the roots [40]. Furthermore, NPA promotes the appearance of roots along the stem of microshoots and far from the wounding site, suggesting a potential role for endogenous hormone in the AR process [80]. Therefore, blocking the auxin movement by NPA is a useful method to analyze other factors involved in AR, as well as to perform expression analysis to confirm the role of the genes putatively involved in the formation of ARs [80].

### 6.4. Ontogenetic Stage

Vegetative propagation of selected genotypes from mature tissues is hindered by the loss of regenerative competence of tissues, as consequence of the chronological, physiological, and ontogenetic ageing that takes place during plant development. Rooting competence in response to auxin is a physiological marker for the ontogenetic state [119] which declines with the transition from the juvenile to adult phase. Phase change (or maturation) is a developmental step that imposes a major shift in the expression of many genes, as seen in *Vitis vinifera* [120], and it is affected by many physiological and environmental conditions [86]. Molecular reasons underlying this shift are believed to rely on epigenetic changes and the activity of microRNAs, as already mentioned [108]. In pea, such decline has been linked to the ontogenetic switch from vegetative to reproductive phase [42].

The juvenile stage is characterized by the high ability to form ARs, among other traits. As shown in Table 2, seedling-derived microshoots exhibited a high rooting ability at the optimal rooting treatments [39,90,95,112]. However, some juvenile clones do not root well (HV-S3) [90], indicating the strong influence of the genetic background on this trait, and thus a breeding program for other traits should also select good rooting ortets.

In woody species, rooting competence is partially lost with maturation. However, the maturation process does not occur at the same time on the entire plant, and some meristems located at the base of the tree retain the ontogenetic juvenile stage [121]. Ontogenetically young shoots that arise from the base of the trunk, like root suckers, stump sprouts, or epicormic shoots produced from pre-existing buds, retain juvenile characteristics while maturation occurs in the periphery of the plant in ontogenetically older but chronologically younger tissues [122,123]. Taking advantage of this cone of juvenility, materials that retain a significant degree of juvenility are frequently chosen for propagating adult trees, when they are available. Most reports dealing with AR in mature chestnuts were developed with these materials and rooting frequencies in microshoots ranged from 19 to 97% (Table 2) [33,90], depending on the clone and rooting treatments. Efforts were also made in chestnut to elucidate the differences in rooting performance in juvenile-like and adult tissues of the same trees, thus avoiding the effect of genotype [90]. The highest rooting frequency achieved in C microshoots was 13.6% [93]. Within each clone, the capacity for AR production was several times higher (3–40) in BS than in C microshoots [92,93]. Although mature microshoots rarely form roots, when it does happen the quality of the root system is very poor with only one or two short and thick roots (Figure 4a). Overall, regardless of the inductive treatment and genotype, microshoots derived from basal shoots root better than crown-derived shoots (Figure 4b).

Grafting has long been used as a propagation method in chestnut. The technique is commonly applied by using juvenile rootstocks, a practice that can improve the rooting rates, while mature scions can be used to maintain the expression of traits of interest. Furthermore, rootstocks that are resistant to soil-borne pathogens, like those that are a major threat to chestnut, can be used. Nonetheless, the cost of this technique is too high for vegetative propagation [5]. Besides, compatibility between scion and rootstock can also diminish the success of the process. Efforts have also been made to increase the in vitro performance of ontogenetically old material by grafting crown sections onto juvenile seedlings [92]. Although the in vitro reactivity was improved, grafting did not affect the rooting response of shoots in any tested clone. In contrast, other authors reported an increase in the rooting rates of microshoots from serial grafting of adult sections onto juvenile rootstocks in *C. sativa*. However, after the stabilization phase, there was a decrease in the rooting capacity, as well as an increase in shoot tip necrosis [95]. AR of mature walnut cuttings was greatly improved by grafting and burying [124]. Micrografting (grafting using microshoots) has also been studied for research purposes. It is a matter of debate what is the real effect of the treatment, if it provides true rejuvenation for the scions or only a certain degree of reinvigoration. Analysis in *C. sativa* microshoots showed that no improvement of the rooting response was achieved by micrografting adult-like material into juvenile-rootstocks, and no rejuvenation signs could be detected in the early stages after grafting [125,126]. Nonetheless, when adult and juvenile-like material with the same genotype where used as scion and rootstock, some reinvigoration/rejuvenation was achieved after several rounds of micropropagation, including a significant increase in rooting rates [127]. Therefore, genotype compatibility might be a key factor for optimization of grafting.

In vitro performance and rooting frequencies of chestnut microshoots derived from mature trees has been improved by spraying the cuttings with 6-benzylaminopurine (BA) during the forced flushing period of new shoots (Table 2) [92]. They also reported an increase in rooting rates by reculturing the explants placed horizontally. By combining ex vitro and in vitro reinvigoration treatments, rooting rates of mature microshoots (clone 431) were similar to those of BS microshoots (Table 2). However, satisfactory rooting rates were not generally achieved with mature tissues despite the efforts in optimizing protocols or using reinvigorating treatments. Besides the genotype influence, the maturation state of the shoots clearly has an enormous influence on the rooting performance of the plants.

### 6.5. Rooting Media

The mineral media most frequently used for the induction of ARs in vitro are Murashige and Skoog (MS) [128] and Gresshoff and Doy (GD) [129], with macronutrients reduced to one half (1/2) or one-third (1/3) strength, respectively, and the usual concentration of sugars. It is believed that such reduction will ease the exit of primordia from the stems as well as help in the growth of new roots, because full-strength media might inhibit rooting, though the response is a species-dependent and probably a genotype-dependent issue. Indeed, rooting rates of microshoots of *C. sativa* and *C. dentata* in the presence of full-strength media were satisfactory [98,112]. In addition, better rooting rates and mean number of roots were achieved in *C. henryi* microcuttings rooted in full-strength media than in half-strength media [100]. The quality of rooted plantlets of the clone Maraval (*C. sativa* × *C. crenata*) was improved by adding ascorbic acid to the rooting medium [130]. Positive results were achieved in microshoots by using mixtures of liquid medium and inert substrates during the expression phase [93,100]. The effectiveness of the Modified Melin-Norkrans (MMN) media has been recently reported on rooting of chestnut microshoots [102].

The use of activated charcoal (AC) is a common strategy for the rooting of cuttings, as it is believed that it will sequester phenolic compounds that might have a detrimental effect on the process, and at the same time provides the dark conditions that will limit the oxidative effect of light on auxins. Low concentrations of AC, ranging from 0.2 to 1%, have been successfully used in the rooting of chestnut microshoots [33,97]. Depending on the genotype, the rooting frequencies were increased up to more than two fold in the presence of AC [93,98]. Nonetheless, as IBA was the hormone frequently used and it is believed to be more stable than IAA in culture media and under light conditions [131], the mode of action of AC in AR remains to be defined. AC has been shown to have contrasting effects on in vitro culture, and its ability to retain plant growth regulators or release stimulatory compounds is not fully characterized [132].

### 6.6. Light Conditions

Light conditions have also been tested in the analysis of AR in chestnut. A period of darkness, usually 3 to 7 days, was applied along with the hormonal treatment during the induction of ARs. In some clones of chestnut, rooting frequencies were enhanced by the darkness period but clonal differences were found [92]. In *C. dentata*, IBA-treated microshoots subjected to an eight-day darkness period rooted better than those subjected to normal photoperiod [97]. The outcome of a dark period during the induction and initiation phases might be related to auxin stability, as previously said, or it might be related to an increase in auxin sensitivity in the tissues. Besides, it is also possible that a darkness period induces the synthesis of endogenous auxin. In a previous work, we showed that a 24 h-dark treatment strongly induced the expression of *CsGH3-1*, a highly auxin-responsive gene, in chestnut microshoots [43]. These results suggest greater synthesis of the hormone or a greater sensitivity of the tissues under dark conditions. In *Petunia hybrida* cuttings, it has been shown that incubation in darkness enhanced the accumulation of endogenous auxin at the stem base [133]. Furthermore, this treatment also accentuates the allocation of carbohydrate resources towards the root forming zone in the same species [134], improving the energy resources needed for the development of the new organ. Further analyses are needed to clarify the effect of a darkness treatment in the auxin homeostasis of tissues and in the rooting process in chestnut.

Overall results suggest that no single protocol can be defined for chestnut species, but rather specific adaptations are needed for every species and genotype, as previously suggested [135]. The use of AC and a period of darkness immediately after dipping, or along with the auxin treatment, apparently have a positive effect, although their application seems not to have straightforward results.

## 7. Root System Architecture and Functionality: In Vitro vs. Ex Vitro Rooting

Root system architecture (RSA), the three-dimensional distribution of the different root components in the soil or growing media, is an essential factor determining the ability of the new roots to explore the soil and supply nutrients for the plant, as well as to provide support for continuous growth. Most research data of AR in chestnut quantifies the root system by the number of roots generated and length of the longest root, which are relevant traits. Nonetheless, a more precise analysis of the RSA is needed to accurately classify the functionality of the new roots. Analysis of the rooting system can be difficult because of the dark nature of substrates or root sensitivity to mechanical damage during manipulation. Secondary roots and root hairs are rarely quantified, mainly because of the lack of precise tools to develop such analysis. New protocols have been developed recently for this purpose, including growing chambers with visualization windows (Rhizotron) or the application of different imaging techniques to phenotype the root (reviewed in [136]). To the best of our knowledge they have not been applied in chestnut.

Data available clearly suggest that ex vitro rooting provides a much stronger rooting system, when compared with in vitro rooting. The use of substrate mixtures for the rooting step also increases the survival rates, though a strong genotype-dependency is found [12]. Gonçalves et al. [33] found that the quality of ex vitro rooted hybrid chestnut shoots was greater than that of in vitro rooted ones, with the former showing the presence of lateral roots and root hairs. In the search for protocol optimization, greater performance of ex vitro rooted plants compared to in vitro rooted plants has also been shown for American chestnut, where the aeration provided by substrates, like vermiculite, improved the formation of lateral roots and root hairs [137]. Indeed, the development of lateral roots in in vitro rooted shoots is rare, and this fact notoriously decreases the subsequent performance of the shoots. The generation of poor rooting systems might increase the mortality rates in the following steps of vegetative propagation.

In vitro plants are subjected to a stressful environment that has an enormous impact on their metabolite profile [138], as well as on different photosynthetic parameters [139], that despite the lack of a proper rooting system might also affect the acclimation success. Indeed, in vitro plants need to develop proper cuticles and epicuticular waxes that help them to stabilize their water apparatus [140]. Besides all these factors, it has been a matter of debate to what extent in vitro generated roots are functional, as the lack of root hairs might render the roots unable to absorb water or nutrients [141]. Fortunately, the plasticity of plants allows them to adapt their rooting system to the new conditions and after few weeks they seem to adapt properly to the new conditions. On the other hand, ex vitro rooting, when successful, speeds up the acclimation process and eases the hardening of in vitro generated plants. It has recently been shown that urea-derivatives can have a positive effect on the development of lateral roots in chestnut in vitro microshoots, and might be related to a better auxin/cytokinin homeostasis during AR development [91].

## 8. Acclimation of In Vitro Rooted Microshoots

Rooted plantlets of chestnut often fail to survive adaptation to ex vitro conditions [12,33,98,139,142]. As rooting and acclimation ability appears to be genotype-dependent [12], the commercial application of current protocols in *Castanea* species and hybrids is seriously limited. Acclimation conditions for chestnut plantlets require fog and mist systems to minimize water loss while most stomata are not completely functional [89,143]. For this process to be successful, it requires a high relative humidity (RH), which is progressively lowered to allow the plants to adapt to the new conditions, enabling them to develop a functional water balance [33,100], which usually coincides with the production of new leaves. Irradiance at about 300 µmol m^−2^ s^−1^ and CO_2_ levels of 700 ppm can improve the survival and growth of the rooted shoots, due to the promotion of the autotrophic behavior [142].

Successful plantlet acclimation depends on the plant material rapidly transitioning from a heterotrophic to an autotrophic habitat at the same time as roots begin supplying water to leaves that may not have developed control of water loss until new leaves and functional stomata develop. For these reasons, efforts to improve chestnut acclimation should focus on the production of functional roots and shoots, as well as defining an optimal environment. Autotrophy can be attained during in vitro culture by using bioreactors with liquid medium and forced ventilation [144]. Temporary or Continuous Immersion Systems (TIS, CIS) were successfully used to proliferate eight ink-resistant *C. sativa* × *C. crenata* genotypes [145,146]. The proportion of rootable shoots (tall, vigorous and with an actively growing apex) suitable to undergo the rooting process was increased, and subsequent acclimation produced good plant survival [147]. Protocols developed for rooting of chestnut include in vitro and ex vitro approaches (Table 3).

When root induction and expression phases are developed in vitro, a subsequent phase of hardening is required, while in the ex vitro approach rooting and acclimation may occur at the same time. When both methods were applied to the same genotypes, survival rates of the latter clearly outperformed the former [33,137]. This might be linked to a more progressive transition to the autotrophic metabolism and to the attaining of functionality of the water balance system, together with a better development of the roots. As outlined in the former section, ex vitro formed roots show a better architecture, with more lateral roots and root hairs than in vitro roots. For ex vitro rooting, disease–free soilless materials with suitable aeration and water-holding characteristics, including mixtures of peat, vermiculite and perlite, can be used, together with other substrates as porous plugs made of cellulose, rockwool or peat, either mixed with bark and plant-derived bio-degradable polymers or surrounded by fine netting. The use of fibrous or porous support materials was shown to be beneficial for difficult-to-root woody plants [148], and in chestnut, they provided good results in hybrids of European and Asian chestnut [146], as well as in American chestnut [149]. Roots developing in the plugs were of better quality than those formed on gelled medium, and the plants required much less handling to transfer them to ex vitro conditions. Transplanting plants along with the support material reduces the possibility of root damage and enables automated handling as in plug seedling production [148].

## 9. Future Prospects of Biotechnological Approaches on AR

Climate change effects, which include higher mean annual temperatures, prolonged periods of drought and the normalization of extreme weather episodes (including flooding events), is one of the greater menaces for humankind in its recent history. Agriculture, besides confronting the challenge of feeding a constantly increasing world population, is one of the more sensitive human activities to the changing environment. The need for a better understanding and an improved use of our plant resources is mandatory if we are to survive as a species. The advancement in our knowledge of AR will ease the path for optimized protocols in vegetative propagation of interesting species.

Tree species of the genus *Castanea* have an enormous relevance both at the ecological and the economical level, but biotechnological tools applied to other trees and crop species have not been readily improved or applied so far. If we want to preserve the natural variability of this genus and take better advantage of the possible benefits it might provide, greater efforts, both scientifically and economically, need to be made.

In this context, micropropagation in photoautotrophic conditions in bioreactors can improve rooting and acclimation of recalcitrant tree species, as well as reduce propagation costs and agar-based residues [144,147]. The application of photoautotrophic conditions (high light intensity, CO_2_ supplementation, and sugar-free medium) to the multiplication and rooting phase of chestnut has been reported for some genotypes [150]. Although all the micropropagation processes can be conducted in this way, the effect is especially remarkable during the rooting process [151]. However, the parameters influencing photoautotrophic propagation at the proliferation stage, including the dynamics of CO_2_ utilization and the overall aeration regime, should be developed more deeply before being applied on a commercial scale.

The transgenic plants developed in chestnut are mainly focused on the generation of genotypes that are resistant to the different threats for chestnut, particularly the pathogens *P. cinnamomi* and *C. parasitica*. To our knowledge, no transgenic plants have been generated in order to improve the rooting performance of chestnut trees. In this sense CRISPR/Cas9, a new genome-editing system, has gained great popularity within the last years in molecular research, as well as in plants. The protocol provides for efficient targeted modifications of specific sequences within the genome, thus opening the possibility to improve quality of the plants, biotic, and abiotic responses and fine-tuning expression of the genes, etc. [152]. Its application, together with other tools, should pave the way for an improved fitness of crops and trees.

The availability of complete genomes for this genus should pave the way for a deeper understanding of the reasons that underlie the genotype differences that are found to be the reason behind the differential responses to the rooting treatments in the works analyzed here. Chinese chestnut is on the verge of a significant biotechnological burst due to the completion of the sequencing of its genome and the development of a transformation protocol [153,154]. Together with the work developed by the Hardwood Genomics Initiative in chestnut (hardwoodgenomics.org), available information might be crucial for the development of biotechnological advances that will improve our knowledge concerning this genus, as well as the development of protocols that might help overcome the difficulties in its improvement. Furthermore, more molecular analyses are needed in order to develop the biotechnological approaches that will enable the scientific community to overcome the main limitations in vegetative propagation of chestnut, as it is clear that biotechnology will ultimately provide the tools for the improvement in AR demanded by the chestnut production industry. Moreover, the application of protocols that might modify the epigenetic status of adult tissues is an unexplored possibility with potential positive results.

Furthermore, the use of bioinformatics approaches is also growing in the plant biotechnological field, including modeling strategies that use generated data to obtain improved protocols by means of neural network or fuzzy logic. These protocols have been successfully applied to improve in vitro conditions or acclimation processes in other species [155,156]. The application of these approaches in chestnut could also provide significant advances for the biotechnological exploitation of this genus.

Therefore, though relevant advances have been made, there is an urgent need for more research in chestnut, particularly at the molecular and genetics level. The knowledge acquire will pave the way for the understanding of the limitations in vegetative propagation of these species, as well as it will help improve current protocols that will provide a more efficient exploitation of its economic potential and allow for the conservation of the genomic variability of the genus.

## Figures and Tables

**Figure 1 plants-09-01543-f001:**
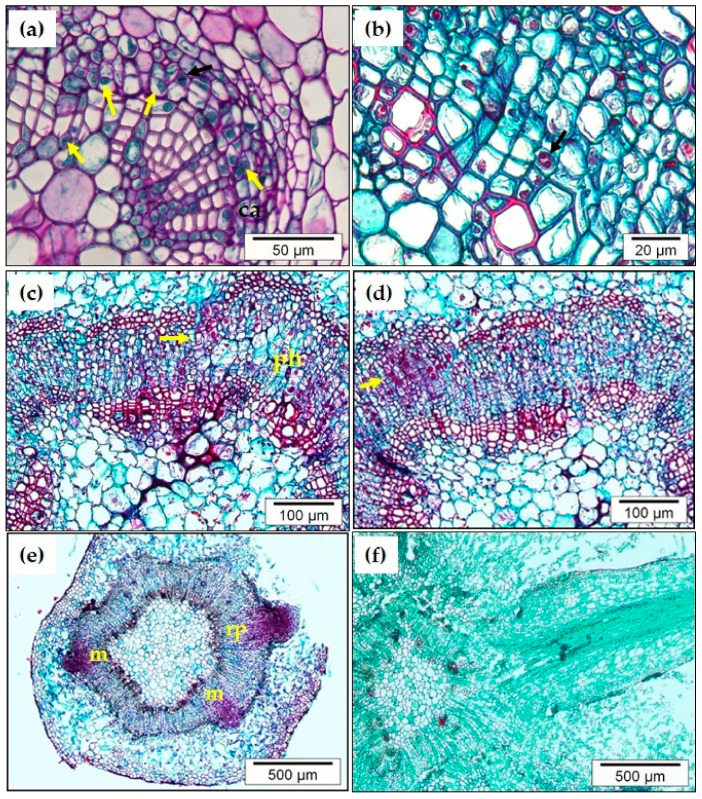
Histological analysis during the adventitious root formation in juvenile-like *C. sativa* microshoots treated with 4.9 mM of indole-3-butyric acid (IBA). (**a**) Periclinal (**a**) and anticlinal (**b**) mitotic divisions (black arrows) in the cambial zone (ca) after 24 h of IBA treatment. (**b**,**c**) Cell activation and cell division increase in the cambial and the phloem (ph) areas (yellow arrows) after 72 h (**c**) of treatment. (**d**) At day 5, clusters of meristematic cells are observed in the outer phloem (yellow arrow). (**e**) Meristemoids (m) and early root primordia (rp) are seen at day 6, which thereafter (**f**) grow, and eventually develop into a functional adventitious root with vascular connection to the stem base.

**Figure 2 plants-09-01543-f002:**
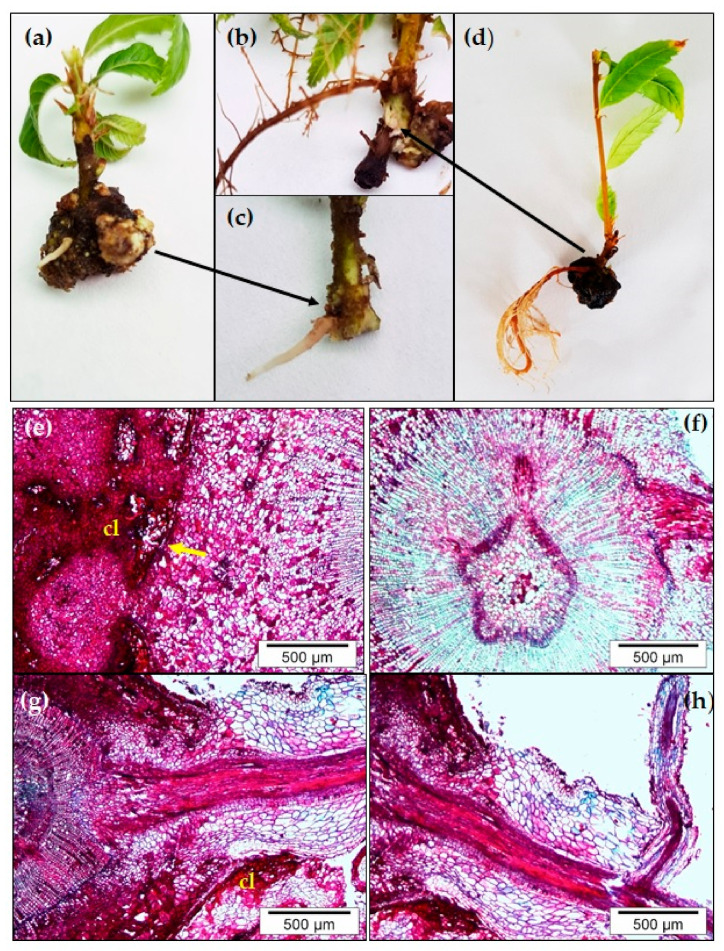
Formation of spontaneous roots in juvenile-like *Castanea sativa* microshoots not treated with auxin. (**a**) Representative images of the callus (cl) developed at the base of the stem of microshoots cultured in the proliferation media for more than two months (**a**,**d**). (**b**,**c**) Close-up images of the base of microshoots shown in (**a**,**b**), respectively, once the surrounding callus was partly removed. (**e**,**f**) Transverse sections of basal parts of microshoots showing (**e**) the callus surrounding the stem (arrow shows the stem epidermis), (**f**–**g**) the connection of root vascular tissue with that of the stem and (**h**) the emerging root with a lateral root.

**Figure 3 plants-09-01543-f003:**
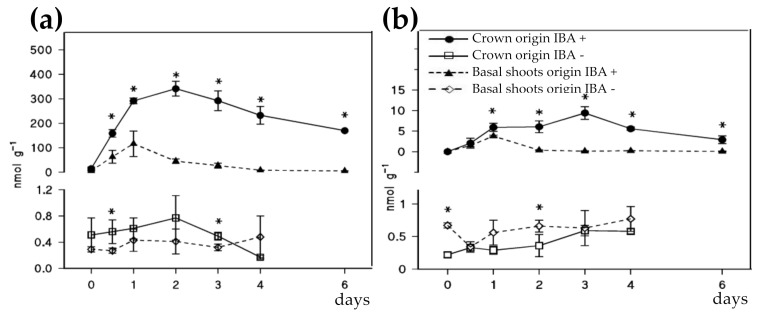
Endogenous indole-3-acetic acid (IAA) content in micropropagated shoots derived from crown branches and basal shoots of clone P1 (*Castanea sativa*) during the first days of rooting. For rooting, shoots were cultured for 24 h in one third strength Gresshoff and Doy medium supplemented with 125 µM of indole-3-butyric acid (IBA+) or not (IBA-), before being transferred to the same medium without auxin. (**a**,**b**) IAA content in the (**a**) basal and (**b**) apical sections of microshoots. Asterisks represent *p* < 0.05 within the same sampling date.

**Figure 4 plants-09-01543-f004:**
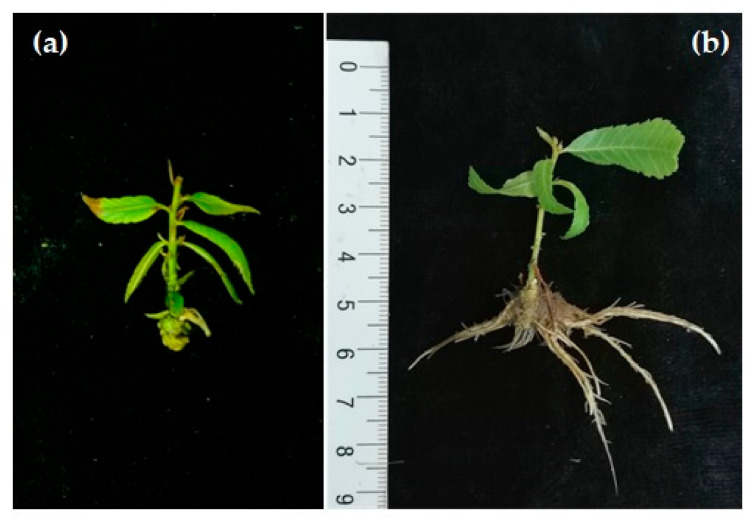
Effect of the ontogenetic stage on the rooting response of *Castanea sativa* microshoots. (**a**) Crown branches-derived microshoots (**b**) Juvenile-like microshoots derived from basal shoots.

**Table 1 plants-09-01543-t001:** Rooting frequencies (RF) of cuttings of *Castanea* genus. Cuttings were taken from new shoots developed from suckers (SS), from stool-bed layering (SB) derived plants or from pruned plants (PS), treated with indole-3-butyric acid (IBA) and placed in different rooting substrates (RS).

Species	Donor Plant	Cutting Type	IBA (mM)	Time (seconds)	RS	RF (%)	Reference
*C. sativa*	1-year-old	PS	10	180–300	Pl	97.0	[10]
*C. dentata*	Adult tress	Semihardwood SS	49	5	P/V	65.0	[89]
	1-yeard old rootstock	Softwood lateral branches	49	5	P/V	59.0	[89]
*C. sativa* × *C. crenata*	6-year-old tree	PS	25	2	P/Sa (3:1)	22.7	[64]
	SB-derived plant	Minicutting	12.3	10	P/Pl (2:1)	33.3	[88]
	SB-derived plant	Minicutting	49	10	P/Pl (2:1)	40.8	[88]
	Ortet 7521	SS	10	180–300	Pl	50.0	[10]
	Ortet 103	SS	10	180–300	Pl	87.0	[10]

P, peat; Pl, perlite; V, vermiculite; Sa, sand.

**Table 2 plants-09-01543-t002:** Rooting frequencies (RF) of chestnut microshoots in response to indole-3-butyric acid (IBA) or naphthalene acetic acid auxin * (NAA), days of darkness (DK) and rooting media (RM).

Species	Clone	Explant Source	Auxin		DK	RM	R (%)	Reference
			(mM)	Time				
*Castanea sativa* (*Cs*)	P1	C	4.9	60 s	0	GD ⅓	8.6	[90]
	P1	BS	4.9	60 s	0	GD ⅓	86.1	[90]
	P1	BS	0.05	24 h	0	GD ⅓	30.5	[91]
	P1	BS	0.125	24 h	0	GD ⅓	94.4	[91]
	P2	C	4.9	60 s	0	GD ⅓	11.4	[90]
	P2	BS	4.9	60 s	0	GD ⅓	94.3	[90]
	P2	BS	0.025	60 s	0	GD ⅓	81.4	[91]
	P2	BS	25	5 d	0	GD ⅓	94.4	[91]
	A2	C	4.9	120 s	0	GD ⅓	0.0	[92]
	A2	C+S	4.9	120 s	0	GD ⅓	18.9	[92]
	A2	BS	4.9	60 s	0	GD ⅓	43.2	[90]
	A3	C	4.9	120 s	0	GD ⅓	8.9	[92]
	A3	C+S	4.9	120 s	0	GD ⅓	22.8	[92]
	L1	BS	0.125	24 h	0	GD ⅓	34.7	[93]
	L1	BS	0.125	24 h	0	GD ⅓ + P/Pl/V	91.7	[93]
	L2	BS	0.125	24 h	0	GD ⅓	20.4	[93]
	L2	BS	0.125	24 h	0	GD ⅓ + P/Pl/V	54.7	[93]
	L4	C	0.250	24 h	0	GD ⅓ + P/Pl/V	13.6	[93]
	Greek	BS	2.7 *	56 d	0	MS ½ + V	56.3	[94]
	Greek	BS	5.4 *	56 d	0	MS ½ + V	50.0	[94]
	Monte	SP	0.015	5 d	5	WPM ½	90.0	[95]
	Monte	C+G	0.015	5 d	5	WPM ½	77.0	[95]
	Pistol	CP	0.015	5 d	5	WPM ½	75.0	[39]
	Pistol	SP	0.02	42 d	0	MS	81.2	[39]
*C. dentata* (*Cs*)	B’ville	SS	5	60 s	0	MS ½ + AC	71.0	[96]
	Iow #2	SP	10	60 s	0	MS ½ + AC	73.0	[96]
	El#1W	SE	10	120 s	0	MS ½ + AC	67.0	[97]
	El#1W	SE	10	120 s	8	MS ½ + AC	89.0	[97]
	El#1	SE	10	30 s	4	WPM + HA	33.3	[98]
	El#1	SE	10	30 s	4	WPM + HA + AC	83.0	[98]
*C. crenata* (*Cc*)	Tanza	SP	0.015	5 d	5	½BW + GG	83.0	[99]
*C. mollisima* (*Cm*)	Yansh	SP	0.015	5 d	5	MS ½ (½NO3)	73.3	[55]
*C. henryi* (*Ch*)	Huali	Se	0.025	28 d	0	MS + Pl	23.3	[100]
	Huali	Se	0.075	28 d	0	MS + Pl	76.7	[100]
	Huali	Se	0.075 *	28 d	0	MS + Pl	0.0	[100]
*Cs* × *Cc*	HV	C	4.9	60 s	0	GD ⅓	0.0	[90]
	HV	BS	4.9	60 s	0	GD ⅓	19.4	[90]
	HV-S3	Se	4.9	60 s	0	GD ⅓	33.3	[90]
	HV-S1	Se	4.9	60 s	0	GD ⅓	100	[90]
	431	C	4.9	120 s	0	GD ⅓	16.9	[92]
	431	C+S	4.9	120 s	0	GD ⅓	30.4	[92]
	431	C+S+R	4.9	120 s	0	GD ⅓	51.0	[90]
	431	BS	4.9	60 s	0	GD ⅓	51.4	[90]
	431-S1	Se	4.9	30 s	0	GD ⅓	97.1	[90]
	110	BS	0.015	7 d	5	GD ⅓	83.0	[93]
	125	BS	0.015	7 d	5	GD ⅓	78.0	[93]
	90025	BS	0.015	7 d	5	GD ⅓	25.0	[93]
	90025	BS	0.25	24 h	5	GD ⅓ + AC	26.0	[93]
	Pr14	BS	0.125	24 h	5	GD ⅓ + P/Pl/V	62.5	[93]
	M3	SS	4.9	60 s	0	MS ½ + AC	97.0	[33]
	M3	SS	0.015	5 d	0	MS ½ + AC	93.0	[33]
	M3	SS	0.015	5 d	0	MS ½	93.0	[101]
*Cs* × *Cm*	SM904	SS	10	60 s	0	MMN	90.0	[102]

AC, activated charcoal; BS, basal shoos; BW, Sato’s BW medium; C, crown branches; C+G, crown branch scion grafted in juvenile seedling; C+G+S, cytokinin spray of shoots developed from C grafted scions; CP, cotyledon petioles; d, day; GD, Gresshoff and Doy medium; GG, gellan gum; h, hour; H, humic acid; MMN, Modified Melin-Norkrans medium; MS, Murashige and Skoog medium; P, peat; Pl, perlite; R, reculturing; S, cytokinin spray of new shoots; s, second; Se, seed; Sa, sand; SE, somatic embryos; SP, seedling plantlet; SS, stump sprouts; V, vermiculite; WPM, Woody plant medium.

**Table 3 plants-09-01543-t003:** Survival rates of micropropagated plantlets of the *Castanea* (C) genus. Root induction was performed under in vitro or/and ex vitro conditions and thereafter microshoots were transferred to different conditions and acclimation periods.

Species	Acclimation Period	Acclimation Conditions	Root Induction	Survival Rate (%)	Reference
*C. crenata*	30 days	Vermiculite + BW ½ medium Continuous light	In Vitro	85	[100]
*C. dentata*	8 weeks	Relative humidity (RH): 80% Fafard’s mix	In Vitro	95	[98]
	8 weeks	RH: 92%; Fafard’s mix	In Vitro/ ex vitro	20/87	[138]
*C. henryi*	90 days	RH: 80% → 65% Peat:perlite (2:1)+MN medium (30 days) Peat: perlite: loess (1:1:1) (60 days)	In Vitro	80	[101]
*C. sativa* × *C. crenata*	4 weeks	RH: 95% → 50%; Peat:perlite (1:2)	In Vitro/ ex vitro	50/100	[33]
	42 days	RH: 98% → ambient; Peat:perlite (1:1)	In Vitro	85	[143]
	8 weeks	RH: 90%; Rockwool cubes (4weeks) Peat:perlite (3:1)(4weeks)	In Vitro	73	[147]

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
