# Peer review of "Recent Advances in Adventitious Root Formation in Chestnut"

_plants, 2020, doi:10.3390/plants9111543_

Round 1

Reviewer 1 Report

Review entitled “Recent advances in Adventitious root formation in chestnut” by Vielba and colleagues summarizes work that have been done in the area of adventitious root formation in chestnut and how current protocols could be improved.

Here are some recommendations:

Page 1 line 19 – sentence could end “including genotype, auxin treatment, light regime and rooting media”. Please remove “etc”.

Page 2 line 83 – remove “etc” and modify sentence to “types of stress such as wounding and flooding [14].

Page 6 line227-233 – in this section the authors mention about auxin accumulation at the wounding zone and auxin gradient. Could there be a similarity not only to lateral development which is later mention in the review but also related to nodule formation where auxin gradients and accumulations are also involved.

Leandro et al 2014 https://www.frontiersin.org/article/10.3389/fpls.2014.00399 

Perrine-Walker et al. (2011). Symbiotic signaling in actinorhizal symbioses. Curr. Protein Pept. Sci. 12, 156–164. doi: 10.2174/138920311795684896

Perrine-Walker et al. (2010). Auxin carriers localization drives auxin accumulation in plant cells infected by Frankia in Casuarina glauca actinorhizal nodules. Plant Physiol. 154, 1372–1380. doi: 10.1104/pp.110.163394

Page 6 Figure 2 (h) the letter is not clear to the dark background. Could the arrows be added to point out the emerging root of the connection in the (h) as it is not clear.

Page 7 line 258- check spelling of “previously”.

Page 10 line 424-426- the sentence is confusing. Is it through their effect of expression?

Page 10 line 430-433 The sentence is too long. May need to modify this sentence.

Page 11 line 451 Spelling of “established”

Page 11 line 459 “as well as”?

Page 13 line 483 “these frequencies were quite stable? Or maintained were quite stable?”

Page 14 section 6.3 – there is also a recent review by Ada and Enrico 2020 on the topic and what about the link to cytokinin?

Page 24 line 1049 chestnut no capital letters Also check if the journal is correct Forests or Forest as mention in reference [106].

Page 26 line 1138 – “in vitro” should be in italics

Author Response

Please see the attachment (response to reviewer 1)

Reviewer 2 Report

Recent advances in adventitious root formation in chestnut

This is a comprehensive and important review on chestnut propagation, written by one of the leading experts in the field and essentially can become “the bible” of chestnut vegetative propagation when published. Nevertheless, there are some issues that to my opinion, should be verified, mainly in respect to the discussions concerning differentiation.

Major concerns:

There are several prominent inaccuracies in definitions in relation to the common literature, for example the use of “dedifferentiation” term, which might be rephrased throughout the text in order to meet the consensus in the scientific community. There are also some inaccuracies in citations. Here are some examples:

  1. “Dedifferentiation is the process whereby the responsive cells remove their old cell fate and revert to a less differentiated state, and gain root competence before their conversion to root initials”.

Dedifferentiation is the process whereby cells revert to their previous status, (please see Regeneration in plants and animals: dedifferentiation, transdifferentiation, or just differentiation? Kaoru Sugimoto, Sean P. Gordon and Elliot M. Meyerowitz, Trends in Cell Biology, April 2011). Therefore, a true example of dedifferentiation process in plants might be related to the ability of plant cells to undergo somatic embryogenesis in which somatic cells revert to embryos in culture. However roots differentiation from cambium stem cells likely do not go through a dedifferentiation step according to this definition.

  1. “The existence of this phase is directly related to the differentiation state of the cells, as they have to go back to a less specialized state in order to increase their developmental potential and be able to respond to the auxin stimulus”.

Please see the previous comment and in addition, auxin signals were shown to be the primary step that define founder cells (J. G. Dubrovsky et al., Auxin acts as a local morphogenetic trigger to specify lateral  root founder cells. Proc. Natl. Acad. Sci. U.S.A. 105, 8790–8794 (2008). Thus, the response to auxin precedes differentiation.

  1. “In any case, dedifferentiation lasts for a few hours, preceding the rooting directed response to auxin. Application of exogenous auxin is generally needed in woody cuttings, and the build-up of the hormone will allow for the priming of root initial cells”.

*Please provide the reference for the estimation of a few hours. In addition, the two sentences contradict each other; while the first one claims that the change in cells precedes auxin perception, the second claims that auxin leads to the priming of root founder cells. Please clarify.

  1. “Essentially for the success of the process, the first division must be asymmetric and switch the cellular division plane, establishing the polarity of the growing root (Citing Druege et al. 2016)”.

 At least in the first cell division during lateral root formation there is no switch in division plane (please see www.pnas.org/cgi/doi/10.1073/pnas.2006387117 ), but if there is a report about adventitious root it should be cited. The Druege review from 2016 does not discuss orientation of initial cell division.

  1. “It is generally accepted that low auxin levels are required for the initiation phase, while higher levels are again required during the expression phase”

It seems that the opposite is true. The paper cited here by the author shows higher auxin pick initially and after a dramatic reduction an additional slight increase. Nag et al J. Plant Growth Regul. 2001, 20, 182–194

.

  1. “Recent studies concerning plant regeneration have suggested that other mechanisms of regeneration share components of the root  developmental pathways, including callus, at least in their first steps [43]. Therefore, at the first steps  of dedifferentiation mature shoots may express the same genetic routes as root-forming shoots but  other factors, physiological and epigenetics impede the establishment of the root developmental  program as those cells do not acquire the rooting competence”.

Not clear. In addition, citation 43 refers to the case in which callus was found to express  root markers even if it was induced from shoots. However,  this callus was generated on callus inducing media in tissue culture. Later it was shown that callus induced by wounding does not express these markers (please see figure 1 in The Plant Cell, Vol. 25: 3159–3173, September 2013 ). The callus developed at the bottom of cuttings likely resemble the latter.

  1. “Another possibility is that some cells retain a memory of their old “stem cell status” and under certain conditions they become root initials”

Not clear, please provide the evidence for the memory. Stem cells are cells with the ability of self-renewal and to give rise to different cells by differentiation. The term de-differentiation does not refer to stem cells. Cambium cells are considered adult stem cells and lead to regeneration of adventitious roots in many plants. (Again, please see Regeneration in plants and animals: dedifferentiation, transdifferentiation, or just differentiation? Kaoru Sugimoto, Sean P. Gordon and Elliot M. Meyerowitz, Trends in Cell Biology, April 2011). In the histological analysis presented here it seems that in chestnut too the cambium is a good candidate for providing the root founder cells.  If the authors want to raise the possibility that adventitious root can differentiate from non-cambium cells, it is again not de-differentiation because for example, parenchyma cells do not revert to their previous status by their convergence to root founder cells. Thus, please consider to rephrase this paragraph.

  1. “At the genetics level, AR formation in chestnut cuttings is believed to be under moderate genetic control” [10], vs  “The great genetic variability that exists within and among species of Castanea greatly influences the rooting performance of the trees”. 

Please explain. How these two sentences co-exist or whether contradict each other.

  1. Lines 191-193: “Molecular mechanisms involved in the emergence of the new ARs have not been explored in chestnut, but it is believed that they mainly resemble those of the lateral roots in other species, with expansin playing a significant role in the advance of the root primordia through the surrounding tissues [38]”.

EXPANSIN A1 (EXPA1), was postulated to modulate the mechanical properties of the pericycle cell wall, being required for radial expansion of lateral root founder cells and to ensure the correct positioning of the first anticlinal divisions (Ramakrishna, P. et al. (2019) EXPANSIN A1-mediated radial swelling of pericycle cells positions anticlinal cell divisions during lateral root initiation. Proc. Natl. Acad. Sci. U. S. A. 116, 8597–8602 ). The reference cited does not provide evidence for expansin’s role in root emergence. 

Minor concerns:

  1. Legend to Figure 1: (c b) arrows;

*There is only one arrow

  1. Line 168: nucleoli become more pronounced:

*It is hard to see nucleoli in this magnification. Please provide higher magnification in an inset.

  1. Lines 169-171 “Whether these histological traits relate to a truly dedifferentiation step is not clear, as they might be related to a more intense transcriptomic activity connected to a cellular reprogramming”:

*Not clear, it seems that the origin of the dividing cells is the cambium which is a stem cells niche, please explain in line with a coherent definition of dedifferentiation and stem cells.

  1. Lines 178-179: After the dedifferentiation step, target cells become sensitive to the auxin stimulus and periclinal mitotic divisions are already detected after 24 hours of indole-3- butyric acid (IBA) treatment in the cambium zone (Figure 1a).

 *Please consider to modify in light of the comments above. First auxin perception and then founder cell formation.

  1. Figure 3: graphs

* What are the units in the X axis

  1. Lines 226-227: Therefore, it appears that the AR initiation and callus formation could share initial genetic pathways as reported by Liu et al. [47] during de novo root regeneration in leaf explants, but then followed a different development.

*In contrast, in another work it was described that a root can be formed directly from the callus (Vegetative propagation of elite Eucalyptus clones as food source for honeybees (Apis mellifera); adventitious roots versus callus formation; Eliyahu et al., 2020 Isr. J. Plant Sci.), suggesting that callus/AR relationships are complex.

This point might be added.

  1. Lines 261-262: These results contrast with those reported in Eucalyptus globulus [50, 51] and Arabidopsis (reviewed in[16]), in which a poor rooting ability was directly correlated to a low auxin content.

*Also in Eucalyptus grandis Abu-Abied et al 2012 The plant Journal.

Lower IAA concentration was found in mature difficult to root cuttings in comparison to juvenile easy to root ones. This reference might be added

  1. Lines 271-277:

* Which two IBA treatments?

  1. Line 316: “Enzymatic activity” a title of new paragraph

*This might be a too broad title, please consider to be more specific

  1. Line 339: Genetics:

*While in the “enzymatic activity” chapter it is written “Nonetheless, these studies might not be conclusive, as the putative existence of isoenzymatic forms or several closely related genes can render the analysis incomplete”.  In the “genetics” chapter there are interpretations on data based on a single transcript analysis belonging to large gene families for example: MYB, SCL etc.

Please explain or add a similar comment at the end of the genetics chapter

  1. Lines361-362 “The α/β hydrolase- like gene showed greater response to auxin in juvenile-like shoots, but lack of characterization of the activity of this putative enzyme prevents us from any conclusions”.

* Xyloglucan endotransglucosylases/hydrolases are highly expressed in Arabidopsis shoot apical meristem (Armezzani, A et al. Development 2018, 145, ) suggesting that reduction in xyloglucan is important to maintain meristematic activity. In agreement, in a paper submitted to this special issue Duman et al. show enrichment in expression of Xyloglucan endotransglucosylases/hydrolases in etiolated cuttings from an Avocado rootstock. In addition, the Arabidopsis xx1/xx2 mutant, hampered in xyloglucan synthesis made more adventitious roots (Duman et al. The contribution of cell wall remodeling and signaling to lateral organs formation. Israeli Journal of Plant Sciences 2020, 67, 110, doi:https://doi.org/10.1163/22238980-20191115.), suggesting a connection between increased expression of hydrolases and AR formation.

This connection might be added.

  1. Lines 425-427; It has been shown in Arabidopsis that the activity of miRNA156 and miRNA157 is directly linked with the vegetative phase change, through their effect the expression of Squamosa Promoter Binding protein/SBP-like (SBP/SPL) transcription factors [80]. Indeed, overexpression of miRNA156 delays transition from the juvenile to the adult [81

*-It may be specified that the latter was shown in a transgenic poplar tree. In Eucalyptus grandis and Eucalyptus brachyphylla it was found that the loss of rooting capability preceded the complete switch between miR156 to miR172 (Levy et al 2014 BMC 25;15(1):524. doi: 10.1186/1471-2164-15-524) and later it was further verified that indeed, a threshold-dependent repression of SPL gene expression by miR156/miR157 is what controls vegetative phase change in Arabidopsis thaliana PLOS Genet 2018 Apr 19;14(4):e1007337. doi: 10.1371/journal.pgen.1007337.

This update may be added.

  1. Lines 566-567: Phase change (or maturation) is a developmental step that imposes a major shift in the expression of many genes, as seen in Vitis vinifera [119], and it is affected by many physiological and environmental conditions [43].

*Citation 43 seems not to be related

  1. Lines 569-570: In sunflower, such decline has been linked to the ontogenetic switch from vegetative to reproductive phase [44].

*Citation 44 is about pea, not sunflower.

  1. Line 642: MNN

*Please indicate abbreviation of what

  1. Lines 40-43 Two repetitive sentences.
  2. Lines 87-88: “Though a strong knowledge of the physiological and molecular basis of the process has been collected..”

*Sentence may be rephrased;

  1. Line 484: excided

*Should be excised?

Author Response

Thank you for you comments and suggestions which help us to greatly improve the manuscript.

Please see the attachement  "response to reviewer 2"

Reviewer 3 Report

The manuscript „Recent advances in adventitious formation in chestnut“ summarizes available data for chestnut propagation (mostly from applied experiments) and places them in the context of plant physiology (e.g. regulatory mechanisms described in other plant species and model plants). It is written with appropriate formal standard. The topic is interesting although rather specialized.

My main complaint about the text is that it is sometimes vague and the information is repetitive in the text (e.g. effects of auxin on chestnut rooting are described several times). I would suggest shorten the text to make it more consistent and easier to read. I have also minor comments that are listed below. In summary, I consider it acceptable after minor revision.

Detailed comments:

Fig.1: The type and concentration of auxin used in the experiment should be specified.

Fig2: The quality of the picture/section h is poor.

Line254: Correct índole to indole.

Line 307: The reference 30 seems incorrect. There is no information about tannins of flavonols in that article.

Line 335: I would suggest to change “conjugating genes” to “auxin conjugating genes”.

Line 389: Arabis alpina is the correct name of the species. Same should be corrected in the references.

Line 453: Experiments with pistachio are placed among results from chestnuts without any information about the relationship of the species. This seems to me confusing.

Table 2: Several abbreviations are not explained (e.g. BW, GD, MS, MMN..)

Line 541: Explain abbreviation MDPU.

Round 2

Reviewer 2 Report

All my comments were nicely addressed